# Support for, and perceived effectiveness of, the UK soft drinks industry levy among UK adults: cross-sectional analysis of the International Food Policy Study

David Pell,[1] Tarra Penney,[1] David Hammond,[2] Lana Vanderlee,[2] Martin White,[1] Jean Adams [1]

[1]Centre for Diet and Activity Research, MRC Epidemiology Unit, University of Cambridge, Cambridge, UK
[2]School of Public Health & Health Systems, University of Waterloo, Waterloo, Ontario, Canada

**Correspondence to**
Dr Jean Adams;
jma79@medschl.cam.ac.uk

## ABSTRACT

**Objectives** To answer four questions: What are attitudes, knowledge and social norms around sugar-sweetened beverages (SSBs)? What are current levels of trust in messages on SSBs? What is current support for, and perceived effectiveness of, the UK soft drinks industry levy (SDIL)? What is the association between attitudes, knowledge, social norms, trust, SSB consumption and sociodemographic factors; and support for, and perceived effectiveness of, the SDIL?

**Design** Cross-sectional online survey.

**Setting** UK.

**Participants** UK respondents to the 2017 International Food Policy Study aged 18–64 years who provided information on all variables of interest (n=3104).

**Outcome measures** Self-reported perceived effectiveness of, and support for, the SDIL.

**Results** Most participants supported the SDIL (70%), believed it would be effective (71%), had a positive attitude to SSBs (62%), had knowledge of the link between SSBs and obesity (90%), and trusted messages from health experts (61%), but not those from the food and beverage industry (73%). Nearly half (46%) had negative social norms about drinking SSBs. In adjusted models, older age, non-consumption of SSBs, social norms to not drinks SSBs, knowledge of the link between SSBs and obesity and trust in health expert messages were associated with greater support for the SDIL, whereas having dependent children and trusting messages from the food and beverage industry were associated with less support. In adjusted models, older age was associated with lower perceived effectiveness of the SDIL, whereas social norms to not drink SSBs, negative attitudes to SSBs and trusting messages from health experts and the food and beverage industry were associated with greater perceived effectiveness.

**Conclusions** There was strong support for the SDIL and belief that it would be effective. Those with more 'public health' orientated norms and trust were generally more likely to support the SDIL or believe that it would be effective.

## Strengths and limitations of this study

► We used a large, population representative sample.
► We were careful to present the Soft Drinks Industry Levy as an intervention targeted at manufacturers rather than consumers, with revenues earmarked for health-promotion activities.
► This is a cross-sectional analysis and we cannot be sure of the direction of causation between putative explanatory variables and outcomes.
► While all have strong face validity, we have not explored other aspects of validity or reliability of any of the measures used; in many cases it would be hard to know what the 'gold standard' measure should be.
► A high proportion of participants who completed the survey were included in the analysis, but we do not know the response rate.

finance) announced a Soft Drinks Industry Levy (SDIL) to be implemented in April 2018.[1] The levy is imposed on industries importing or manufacturing sugar-sweetened beverages (SSBs) and includes two 'tiers'. Drinks with ≥8 g of sugar per 100 mL are charged £0.24 per litre and those with ≥5 g but <8 g per 100 mL are charged £0.18 per litre. Alcoholic drinks, milk-based drinks and pure fruit juices are exempt irrespective of sugar content. The Chancellor stated that revenue raised would be spent on school sport and school breakfast clubs. An explicit aim of announcing the levy 2 years in advance of implementation, and defining two levy tiers, was to provide time for manufacturers to reformulate.[1] The nature and intent of the SDIL makes it unique among international SSB taxes.

The success or failure of policy interventions is often the result of actions and

## INTRODUCTION

In his March 2016 Budget Statement, the UK Chancellor of the Exchequer (minister of

reactions by many stakeholders including government, civil society, industry, the health sector and consumers. In particular support for the SDIL may both be influenced by the SDIL and modify its effectiveness. More intrusive public health interventions, like food and tobacco taxes, generally receive lower levels of public support than less intrusive ones, like information giving.[2] Support for hypothetical SSB taxes has been reported to range from 36%–60%.[3–17]

How a public health intervention is framed may also impact how acceptable it is to stakeholders. The SDIL is specifically framed as a levy on manufacturers, rather than consumers, and as a source of revenue for other health-promoting purposes. The importance of framing interventions such that they redefine public health problems has been previously identified.[18] By specifically targeting manufacturers, the SDIL frames excessive SSB consumption, and the resultant health implications, as a problem of drinks manufacturers, rather than consumers. Support for hypothetical food taxes generally increases when it is proposed that the revenue raised would be used for health-promoting purposes.[11 12 19 20] There is some wider evidence that public health messages in general framed in terms of gains, rather than losses, to recipients elicit more positive responses from the public.[21] Clearly stating that the SDIL is not targeted at consumers (and hence implying that consumers should not lose) and that revenues will be used for health promotion (and hence implying that consumers stand to gain) may, therefore, increase positive responses and hence support for it. Previous work has explored differences in support for SSB taxes according to participant sociodemographic characteristics, but findings are not consistent. For example, support has been varyingly reported as higher in younger people,[3 17 22] higher in older people,[19] and not associated with age.[5 10] Associations between support for SSB taxes and both SSB consumption and markers of socioeconomic position are similarly variable.[3 5 8 16 19 22] Fewer studies have explored psychological correlates of support for SSB taxes, such as attitudes, social norms, knowledge and trust. Those who felt that SSBs were a major (but not minor) contributor to childhood obesity in the USA were more likely to support an SSB tax.[5] Although trust in government was not associated with support in either the UK or USA,[10] more favourable assessments of soft drinks companies were associated with lower support in the USA.[22]

One reason for low support for SSB taxes commonly found in qualitative work is low perceived effectiveness of small changes in price.[6 7 11 12 20] Perceived effectiveness is less studied in quantitative studies, but has been found to range from 39%–58%.[5 12 19] Perceived effectiveness was found to be an important correlate of support in one quantitative study,[10] and has also been reported to be higher in older people and those with more education; but lower in those consuming more SSBs.[19]

The great majority of work in this area has focused on hypothetical taxes. As support for more intrusive public health interventions often increases after implementation,[2] support for hypothetical SSB taxes may misrepresent support for taxes that have been announced or implemented. To date, we are aware of only one study that has explored public perceptions of a definite, rather than hypothetical, tax on drinks.[19] This study was conducted in France where an excise tax applies to all sweetened drinks, including those sweetened with artificial sweeteners. Given the difference between the French tax and SSB taxes, which are more specific to drinks sweetened with sugar, the French findings may not be generalisable.

In this study we explored both sociodemographic and psychological correlates of support for, and perceived effectiveness of, a definite, rather than hypothetical, SSB tax that has been framed in a unique way. Using data from UK adults collected 20 months after announcement and 4 months before implementation of the SDIL, our specific research questions were: 1. What are current attitudes, knowledge and social norms around SSBs? 2. What are current levels of trust in messages on SSBs from different institutions? 3. What is current support for, and perceived effectiveness of, the SDIL? 4. What is the association between attitudes, knowledge, social norms, trust, SSB consumption and sociodemographic factors; and support for, and perceived effectiveness of, the SDIL?

## METHODS

The analyses were prespecified in a protocol.

### Sampling, recruitment and data collection

Data were from UK participants in wave 1 of the International Food Policy Study, conducted in Australia, Canada, Mexico, the UK and the USA. Data were collected via self-completed web-based surveys in December 2017 with adults aged 18–64 years. Respondents were recruited through Nielsen Consumer Insights Global Panel and their partners' panels. Email invitations (with a unique link) were sent to a random sample of panellists (after targeting for age and country criteria); panellists known to be ineligible were not invited. The mean survey time across countries was 33 min.

Respondents provided consent prior to completing the survey. Respondents received remuneration in accordance with their panel's usual incentive structure (eg, points-based or monetary rewards, or chances to win prizes). A full description of the study methods can be found in the International Food Policy Study: Technical Report—Wave 1 (2017) at www.foodpolicystudy.com/methods.

### Variables used in the analysis

The variables used in the analysis, the survey items they were derived from, response options and how response options were collapsed for analysis are described in table 1.

Alongside single-item measures of attitudes, knowledge and social norms related to sugary drinks; we included

**Table 1** Description of items and response options used in the analysis

| Concept | Item wording (where applicable) | Response options — All | Used in analysis |
|---|---|---|---|
| Age | How old are you? | In years | In years |
| Sex | What sex were you assigned at birth, meaning on your original birth certificate? | Female | Female |
| | | Male | Male |
| Education | What is the highest level of education you have completed? | Qualifications not listed below, free-text equivalents, Don't Know, Refuse to answer | ≤School leaving |
| | | NVQ Level 4–5, HNC, HND, RSA Higher Diploma, BTEC Higher Level, Degree, Higher Degree, free-text equivalents | >School leaving |
| Income sufficiency | How easy is it to make ends meet? | Neither easy nor difficult, Difficult, Very difficult, Don't know, Refuse to answer | Not easy |
| | | Very easy, Easy | Easy |
| Children | Do you have any children (including step-children or adopted children) under the age of 18? | No, Don't know, Refuse to answer | No |
| | | Yes | Yes |
| SSB consumption | [Calculated from Beverage Frequency Questionnaire: reported consumption over last 7 days] | Any consumption of non-diet Fizzy drinks, Sweetened fruit juice drinks, Regular sports drinks, Regular energy drinks, or Spirits with mixers that have calories | Consumers |
| | | No consumption of above | Non-consumers |
| Social norms | People important to me try not to drink sugary drinks | Neither agree nor disagree, Disagree, Strongly disagree, Don't know, Refuse to answer | Not agree |
| | | Strongly agree, Agree | Agree |
| Attitudes | Sugary drinks taste good | Strongly agree, Agree | Agree |
| | | Neither agree nor disagree, Disagree, Strongly disagree, Don't know, Refuse to answer | Not agree |
| Knowledge | Frequently drinking sugary drinks increases the risk of obesity | False, Don't know, Refuse to answer | Not true |
| | | True | True |
| Expert trust | I trust messages from health experts on sugary drinks | Neither agree nor disagree, Disagree, Strongly disagree, Don't know, Refuse to answer | Not agree |
| | | Strongly agree, Agree | Agree |
| Industry trust | I trust messages from the food and beverage industry on sugary drinks? | Neither agree nor disagree, Disagree, Strongly disagree, Don't know, Refuse to answer | Not agree |
| | | Strongly agree, Agree | Agree |
| Support | In 2018 a new sugary drink tax will be introduced in the UK. This aims to encourage manufacturers to reduce the sugar in drinks. The money will be spent on breakfast clubs, and sports in primary schools. Do you support or oppose this policy? | Strongly support, Support | Support |
| | | Oppose, Strongly oppose, Don't know, Refuse to answer | Oppose |
| Effectiveness | Preamble as above. How effective do you think these kinds of policies are? | Somewhat effective, Mostly effective, Very effective | Effective |
| | | Not at all effective, Don't know, Refuse to answer | Not effective |

single items measures of trust in advice on sugary drinks from health experts, and the food and beverage industry; and single item measures of support for, and perceived effectiveness of, the SDIL. As previous research has indicated that the acceptability of food taxes varies with the stated intentions of these,[11 12 19 20] we included a preamble to the questions about support for, and perceived effectiveness of, the SDIL outlining the intention of the levy and the stated use of revenue generated.

Sociodemographic variables considered were age in years, sex at birth, whether or not participants had children and socioeconomic position. Parental status was a potentially important variable because the SDIL is included as a flagship component of England's Childhood Obesity Plan and has particularly been framed in terms of potential benefits to children.[1 23] Socioeconomic position was measured using participants' highest educational qualification and perceived income sufficiency.

The Beverage Frequency Questionnaire is a 7-day food record that assesses consumption for 17 beverage categories, including caloric and non-caloric beverages.[24] For each beverage category, respondents report the number of drinks and the usual portion size, using category-specific images of beverage containers, adapted from the ASA24 dietary recall.[25] Participants who reported any consumption of regular fizzy drinks (including alcoholic drinks that contained regular fizzy drinks as a mixer), sweetened fruit drinks, sports drinks or energy drinks over the previous 7 days were considered SSB consumers in the analysis.

### Inclusion criteria

UK resident participants in wave 1 of the International Food Policy Study, aged 18–64 years, who correctly responding to a data integrity question in which participants were asked to identify the current month, and provided usable information on all other variables of interest were included in the analysis. Data from countries other than the UK were not included as comparable questions on support for, and perceived effectiveness of, the SDIL were not asked of these participants.

### Analysis

Data were weighted with poststratification sample weights constructed using population estimates from the UK census based on age group, sex and region. These sample weights were used throughout the analysis to reduce the effects of non-response and selection bias and return the sample to population representativeness.

Descriptive statistics were used to quantify all variables of interest. Logistic regression models were fitted to explore associations between other variables and support for, and perceived effectiveness of, the SDIL. We used separate models to explore support for the SDIL and perceived effectiveness of the SDIL where support for, or perceived effectiveness of, the SDIL were the outcome variables and all other variables

were included as explanatory variables. Unless otherwise noted, adjusted OR (and 95% CI) of support for, or perceived effectiveness of, the SDIL are presented adjusted for all other variables included.

Data were analysed using R V.3.3.1.

### Patient and public involvement

Patients and the public were not involved in design, conduct, analysis or interpretation of the study.

### RESULTS

Of 4276 who took part in the in the UK arm of the International Food Policy Study in December 2017, 4047 (95%) correctly responded to the data integrity question. Of these, 3104 (77%) provided complete data on all variables of interest and were included the analysis.

Characteristics of the analytical sample (after applying survey weights) are described in table 2. Participants had a mean age of 38 (SD 13) years, with a good balance across sex at birth (48% female). The highest level of education that most participants had achieved was the equivalent of school-leaving or lower and around two-thirds (61%) did not find it easy to make ends meet. Just over one-third (37%) of participants had children under the age of 18 years, and just less than half (47%) reported consuming SSBs in the last 7 days.

Around half of participants (54%) agreed that people important to them try not to drink sugary drinks (social norms), around two-thirds (62%) that sugary drinks taste good (attitudes), and 90% believed that frequently consuming sugary drinks increases the risk of obesity (knowledge). While more than half (61%) of respondents trusted messages from health experts on sugary drinks, only one-quarter (27%) trusted messages from the food and beverage industry.

Table 3 shows the results of logistic regression analyses of associations between sociodemographics, social norms, attitudes, knowledge and trust, and perceived support for, and effectiveness of, the SDIL—adjusted for all other variables in the models.

In adjusted models, older participants were more likely to support the SDIL, but were less likely to consider it effective. Those with dependent children and those who trusted messages from the food and beverage industry on sugary drinks were less likely to support the SDIL. Non-consumers of SSBs, those with social norms to not drinks sugary drinks, those with knowledge of the association between sugary drinks and obesity, and those who trust messages from health experts on sugary drinks were more likely to support the SDIL than others. Those with high social norms around not drinking sugary drinks, less positive attitudes to sugary drinks, and those who trusted messages on sugary drinks from health experts and from the food and beverage industry were more likely to consider the SDIL would be effective. There were no differences in

**Table 2** Weighted characteristics of UK participants in the International Food Policy Study, Dec 2017 (n=3104)

| Concept | Question wording (where applicable) | Response category | n | % |
|---|---|---|---|---|
| Sex | What sex were you assigned at birth, meaning on your original birth certificate? | Female | 1497 | 48 |
| Education | What is the highest level of education you have completed? | A-Levels or lower | 1896 | 61 |
| Income sufficiency | How easy is it to make ends meet? | Not easy | 1905 | 61 |
| Children | Do you have any children (including step-children or adopted children) under the age of 18? | No | 1963 | 63 |
| SSB consumption | Consumed regular fizzy drinks, sweetened fruit drinks, sports drinks, energy drinks in last 7 days | Consumers | 1473 | 47 |
| Social norms | People important to me try not to drink sugary drinks | Not agree | 1416 | 46 |
| Attitudes | Sugary drinks taste good | Agree | 1938 | 62 |
| Knowledge | Frequently drinking sugary drinks increases the risk of obesity | Not true | 322 | 10 |
| Expert trust | I trust messages from health experts on sugary drinks | Not agree | 1213 | 39 |
| Industry trust | I trust messages from the food and beverage industry on sugary drinks | Not agree | 2267 | 73 |
| Support | In 2018 a new sugary drink tax will be introduced in the UK. This aims to encourage manufacturers to reduce the sugar in drinks. The money will be spent on breakfast clubs, and sports in primary schools. Do you support or oppose this policy? | Support | 2167 | 70 |
| Effectiveness | Preamble as above. How effective do you think these kinds of policies are? | Effective | 2214 | 71 |

support for or perceived effectiveness of the SDIL by sex, education or perceived income sufficiency.

## DISCUSSION
### Summary of findings
To our knowledge, this is the first study of a range of socio-demographic, consumption and psychological correlates of both support for, and perceived effectiveness of, an SSB tax. Unlike previous studies, our research was conducted in the context of a definite, rather than hypothetical, SSB tax. We found that the majority of UK adults aged 18–64 years were supportive of the SDIL and believed it would be effective, have a positive attitude to sugary drinks, have good knowledge about the links between sugary drinks and obesity, and trust messages from health experts, but not the food and beverage industry, about sugary drinks. Around half reported social norms about not drinking sugary drinks.

Social norms towards not consuming sugary drinks and trusting health expert messages on sugary drinks were both associated with greater support for and perceived effectiveness of the SDIL. In addition, having dependent children and trusting messages from the food and beverage industry on sugary drinks were associated with less support for the SDIL, while older age, not consuming SSBs and knowledge of the link between sugary drinks and obesity were associated with greater support. Older age was associated with lower perceived effectiveness of

the SDIL, and more negative attitudes towards sugary drinks were associated with greater perceived effectiveness. There were no associations between gender, education or income sufficiency and either support for, of perceived effectiveness of, the SDIL.

### Strengths and weaknesses of methods
Key strengths of the analysis are the use of a large, population representative, sample; inclusion of a range of socio-demographic, consumption and psychological variables; and the context of a definite, rather than hypothetical, SSB tax announced 20 months before data collection (although not implemented until 4 months after). Given previous findings that support is greater when revenues are used for health-promoting activities,[11 12 19 20] we were careful to present the SDIL as an intervention targeting manufacturers rather than consumers, with revenues ear-marked for health-promotion activities. Social desirability bias may be less likely to occur in more anonymous settings such as online surveys.[26]

Participants were not recruited using probability-based sampling meaning the findings do not provide nationally representative estimates, although this was reduced by applying sampling weights. The results are, therefore, likely to be generalisable to the UK, but may not be more widely generalisable. This is a cross-sectional analysis and we cannot be sure of the direction of causation between putative explanatory variables and outcomes. Nor have we explored more complicated

**Table 3** Adjusted* OR (95% CI) of characteristics associated with support for, and perceived effectiveness of, the SDIL

| Concept | Question wording (where applicable) | Response category | Adjusted OR (95% CI) of SDIL support | Adjusted OR (95% CI) of SDIL effectiveness |
|---|---|---|---|---|
| Age | How old are you? | Years | **1.01 (1.00 to 1.02)** | **0.99 (0.98 to 0.99)** |
| Sex | What sex were you assigned at birth, meaning on your original birth certificate? | Female | Reference | Reference |
| | | Male | (0.72 to 1.05) | 1.03 (0.85 to 1.25) |
| Education | What is the highest level of education you have completed? | A-Levels or lower | Reference | Reference |
| | | >A-Levels | 1.03 (0.85 to 1.26) | 0.90 (0.73 to 1.10) |
| Income sufficiency | How easy is it to make ends meet? | Not easy | Reference | Reference |
| | | Easy | 1.01 (0.83 to 1.24) | 1.02 (0.83 to 1.25) |
| Dependent children | Do you have any children (including step-children or adopted children) under the age of 18? | No | Reference | Reference |
| | | Yes | **0.81 (0.67 to 0.99)** | 1.16 (0.94 to 1.43) |
| SSB consumption | Consumed regular fizzy drinks, sweetened fruit drinks, sports drinks, energy drinks in last 7 days | Consumers | Reference | Reference |
| | | Non-consumers | **1.57 (1.28 to 1.91)** | 1.21 (0.99 to 1.48) |
| Social norms | People important to me try not to drink sugary drinks | Not agree | Reference | Reference |
| | | Agree | **1.39 (1.15 to 1.70)** | **1.25 (1.03 to 1.53)** |
| Attitudes | Sugary drinks taste good | Agree | Reference | Reference |
| | | Not agree | 1.10 (0.89 to 1.36) | **1.31 (1.07 to 1.61)** |
| Knowledge | Frequently drinking sugary drinks increases the risk of obesity | Not true | Reference | Reference |
| | | True | **2.34 (1.74 to 3.16)** | 1.06 (0.77 to 1.45) |
| Expert trust | I trust messages from health experts on sugary drinks | Not agree | Reference | Reference |
| | | Agree | **2.01 (1.63 to 2.49)** | **1.86 (1.51 to 2.28)** |
| Industry trust | I trust messages from the food and beverage industry on sugary drinks | Not agree | Reference | Reference |
| | | Agree | **0.55 (0.44 to 0.69)** | **1.37 (1.08 to 1.75)** |

*All results are adjusted for all other variables listed.
Bold indicates statistically significant at the P<0.05 level.
SDIL, soft drinks industry levy.

causal networks linking the variables included. All variables were self-reported. While all have strong face validity, we have not explored other aspects of validity or reliability of any of the measures used. However, all were derived from existing instruments and in many cases it would be hard to know what a 'gold standard' measure should be for validation. Although a high proportion of participants who completed the survey were included in the analysis, we do not know what proportion of those invited to participate were included.

### Comparison to previous results and interpretation of findings

Most people in our survey (90%) knew that there was an association between sugary drink consumption and obesity. This reflects previous findings where 89%–91% agreed that SSB consumption increased the risk of obesity.[3 5] Despite this, there were also high positive attitudes towards sugary drinks with almost two-thirds of respondents agreeing that sugary drinks taste good, and less than half had social norms about not drinking sugary drinks. In the UK, SSBs appear to remain a pleasurable and positive part of life, despite their known health harms.

Similar to previous research which found that only 30% of Americans gave favourable ratings to soda companies,[22] we found low levels of trust in messages about sugary drinks from the food and beverage industry. Levels of trust in similar messages from health experts were higher, but still less than two-thirds. Low levels of trust in experts may reflect a general public mistrust of nutritional epidemiology.[27]

Despite less than perfect trust in messages about sugary drinks from health experts, there was a high level of support for the SDIL (70%) and even higher belief that it would be effective (71%). This is higher than previous research which, as far as we are aware, reports maximum support of 60%.[8 28] Even in the context of an existing tax on sweetened drinks in France, only 49% supported the tax.[19] The high level of support we found may reflect the combined effect of previous findings that support for more intrusive public health interventions such as taxes on food and tobacco often increases after implementation,[2] and that support for SSB taxes is often greater when revenues are used for health-promoting activities.[11 12 19 20 28 29] Although the SDIL had not been implemented at the time of data collection, impending implementation had been known of for 20 months. Further, we were careful to inform participants that SDIL revenues would be spent on school breakfast clubs and sports activities. In addition, the SDIL is unique in being targeted at manufacturers rather than consumers, and intended to promote reformulation rather than necessarily reduce consumption.[1] Previous qualitative work has found that those who do not support generic SSB taxes often cite excessive personal taxation and government intrusion into individual's lives as reasons for this.[11 30] This is much less applicable to the SDIL than to consumer-facing SSB taxes.

Low acceptability of SSB taxes has previously been ascribed to a perception that they are unlikely to achieve significant behaviour change or public health benefit.[11 30] Previous research has reported perceived effectiveness (to improve population health or decrease SSB consumption) in the range of 39%–58%.[5 12 19] In contrast, we found much higher levels of perceived effectiveness (71%). This may again reflect the unique nature of the SDIL with an explicit intention to change manufacturer, rather than consumer, behaviour— and our focus on effects on industry, rather than consumer, behaviour.

Higher support for, and perceived effectiveness of, the SDIL here compared with previous work may also reflect cultural differences between the UK and other countries where previous data has been collected. Unlike previously, we used population weighting which increases confidence that results are population representative. Finally, it is possible that the unique design and framing of the SDIL makes it more acceptable and increases perceived effectiveness compared with previous taxes proposed to research participants.

The pattern of associations between attitudes, social norms, trust and support for, and perceived effectiveness of, the SDIL are, for the most part, intuitive. It might be expected that non-consumers, who are less likely to be negatively financially effected by the tax, would be more supportive. In other contexts, those who stand to gain most from financial incentive interventions are most supportive.[31] Social norms to not drink sugary drinks, negative attitudes towards sugary drinks, greater knowledge about the health harms of sugary drinks, greater trust in health experts and less trust in the food and beverage industry all reflect more 'public health' orientated patterns that would be expected to be associated with greater support for, or perceived effectiveness, of the SDIL.

It is somewhat surprising that those with children under the age of 18 years were less supportive of the SDIL than those without. The SDIL was particularly framed in terms of potential benefits to children.[1 23] If one's own consumption is likely to influence support for the SDIL, then parents' support for the SDIL may also be influenced by their children's consumption. If children are greater consumers of sugary drinks,[32] then this may explain why parents with children under the age of 18 years were less supportive. As described above, previous research on the association between psychological variables and support for, and perceived effectiveness of, SSB taxes is sparse.

We did not find that gender or markers of socioeconomic position were associated with support for, or perceived effectiveness of, the SDIL in mutually adjusted models. This reflects some, but not all, previous findings.[5 8 10 19 22] Unlike most previous work we included a wide range of sociodemographic, consumption and psychological variables in mutually adjusted models and it may be that gender or socioeconomic differences operate entirely through the other variables included in our models.

## Implications of findings

Many structural public health policies require government action, which may be limited by perceptions concerning public acceptability of such policies—often uninformed by evidence. Greater understanding of public acceptability of a range of structural public health policies, and how this changes over time and over the course of implementation, may help to develop strategies to address public concerns and build public support for these policies.

## CONCLUSIONS

UK adults tend to have positive attitudes to sugary drinks and do not necessarily have strong social norms about not drinking sugary drinks, but they generally recognise the link between sugary drink consumption and obesity. Trust in messages about sugary drinks from the food and drinks industry is low, but trust in these messages from health experts is not universally high. There was strong support for the SDIL and belief that it will be effective. Those with more 'public health' orientated norms and trust were generally more likely to support the SDIL or believe that it will be effective, although those with dependent children were less likely to support the SDIL.

**Contributors** JA, TP and MW conceived the idea for this paper. DP analysed the data. JA drafted the manuscript. JA, TP, MW, DH and LV read and provided critical comments on the manuscript and approved the final version. DH conceived the idea for the IFPS and secured funding. DH and LV developed the first draft of the survey. TP led the further development of the UK survey instrument, with input from JA, MW, DH and LV.

**Funding** Funding for the International Food Policy Study was provided by the Canadian Institutes of Health Research (CIHR; operating grant). Additional support was provided by a CIHR—Public Health agency of Canada (PHAC) Applied Public Health Research Chair. The study has no affiliations with the food industry. The analyses reported in this paper were supported by The Health Foundation. JA and MW are supported by the Centre for Diet and Activity Research (CEDAR), a UKCRC Public Health Research Centre of Excellence. Funding from the British Heart Foundation, Cancer Research UK, Economic and Social Research Council, Medical Research Council, the National Institute for Health Research, and the Wellcome Trust, under the auspices of the UK Clinical Research Collaboration, is gratefully acknowledged (grant number MR/K023187/1). Views expressed in this paper are those of the authors and not necessarily those of the above named funders.

**Competing interests** None declared.

**Patient consent for publication** Not required.

**Ethics approval** The study was reviewed by and received ethics clearance through a University of Waterloo Research Ethics Committee (ORE# 21460).

**Provenance and peer review** Not commissioned; externally peer reviewed.

**Data sharing statement** Data are available directly from the International Food Policy Study team on reasonable request (see www.foodpolicystudy.com).

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
