## [Reviewer comments · BMJ Open]

ARTICLE DETAILS

TITLE (PROVISIONAL)	SUPPORT FOR, AND PERCEIVED EFFECTIVENESS OF, THE UK SOFT DRINKS INDUSTRY LEVY AMONGST UK ADULTS: CROSS-SECTIONAL ANALYSIS OF THE INTERNATIONAL FOOD POLICY SURVEY
AUTHORS	Pell, David; Penney, Tarra; Hammond, D; Vanderlee, Lana; White, Martin; Adams, J

VERSION 1 – REVIEW

REVIEWER	Frederick Zimmerman UCLA, USA
REVIEW RETURNED	05-Oct-2018

GENERAL COMMENTS	This is a strong piece of work and for the most part very well written up. It makes a vital contribution to the literature. My only major comment is that the contribution it makes isn't highlighted as much as it should be. I would recommend adding another 1-3 paragraphs about the value of framing the reasons for a tax, using credible messengers, tying the revenue from a tax to health-promoting expenses, and so on. There is a little bit of work on framing out there, but not much. The paper should be structured so as to centralize its practical value to policy-makers looking to sell a sugar-sweetened beverage tax. Minor comments 5 strengths and limitations of the study are listed. The one about not being able to draw causal inferences can be dropped, because this is a purely descriptive paper. p. 5 line 12: it isn't clear what is meant by a "real" tax. Is this just a tax that has already taken effect? If so, so state. (See also top of p. 12) p. 5, line 39: The questions specifically about the SDIL could be asked only of the UK participants. Were similar questions asked of participants in other countries? If so, are any comparisons possible? In Table 2, I don't see the value in reporting both the yes and no percentages for binary questions. They seem always to sum to 100%. However, it might be useful to report both the raw percentages and the percentages adjusted for the survey weights. In Table 3 I would rather not see the n and (%) for the binary variables, because the methods for this calculation are not
--

	reported and anyway, the adjusted odds-ratio is what we're here for. In Table 3, I worry that there is some spurious significance caused by the relatively large number of variables included, without strong a priori reasons to distinguish expected causal relationships. I would suggest using a Bonferroni correction to the statistical significance. That will result in fewer variables being significant, but a good deal more confidence in the ones that are significant. In the discussion section, one reason these results may differ from others is the large sample size here, coupled with the use of population weights. It is possible that some of the previous results may have come from smaller and more selected samples. If so, this is a point worth making in some detail. Another possible difference is the threshold structure of the SDIL—I doubt that this is really driving the differences, but if the authors' expertise can shed any light on this issue it would be helpful.
--	---

REVIEWER	Dr. Orly Tamir Israeli Center for Diabetes Research and Policy, The Gertner Institute, Israel
REVIEW RETURNED	02-Nov-2018

GENERAL COMMENTS	Manuscript is well written and organized. However, authors need to respond to a few issues:  1. I suggest that the authors relate to tax on tobacco - an area which 'real-tax' was studied. This should be added to both Introduction and Discussion sections. 2. Methods section - Please provide the weights that were used throughout the analysis and refer to their source. Also, add information on the software that was used to analyse the data, and add the level of significance. 3. Discussion - I am missing a discussion over the notable result that respondents with young children were less likely to support the tax. If we (the society in whole) aim to improve dietary habits of youngsters, it might be difficult if the parents have a negative attitude towards the tax. I would relate to this point also in the Conclusions or Implications i.e. disseminate information to the public on this regulatory measure.
---

VERSION 1 – AUTHOR RESPONSE

In response to Reviewer #1

My only major comment is that the contribution it makes isn't highlighted as much as it should be. I would recommend adding another 1-3 paragraphs about the value of framing the reasons for a tax, using credible messengers, tying the revenue from a tax to health-promoting expenses, and so on. There is a little bit of work on framing out there, but not much. The paper should be structured so as to centralize its practical value to policy-makers looking to sell a sugar-sweetened beverage tax.

Thank you for proposing this different way of 'framing' our research. We have added a substantial extra paragraph to the introduction stating that: "How a public health intervention is framed may also impact how acceptable it is to stakeholders. The SDIL is specifically framed as a levy on

manufacturers, rather than consumers, and as a source of revenue for other health promoting purposes. The importance of framing interventions such that they redefine public health problems has been previously identified. By specifically targeting manufacturers, the SDIL frames excessive SSB consumption, and the resultant health implications, as a problem of drinks manufacturers, rather than consumers. Support for hypothetical food taxes generally increases when it is proposed that the revenue raised would be used for health promoting purposes. There is some wider evidence that public health messages in general framed in terms of gains, rather than losses, to recipients elicit more positive responses from the public. Clearly stating that the SDIL is not targeted at consumers (and hence implying that consumers should not lose) and that revenues will be used for health promotion (and hence implying that consumers stand to gain) may, therefore, increase positive responses and hence support for it." Page 4, par 4.

5 strengths and limitations of the study are listed. The one about not being able to draw causal inferences can be dropped, because this is a purely descriptive paper.

Given the reviewer's concern (below) that we may not have confidence in the data to identify causal relationships, we have opted to retain this limitation. It clarifies very clearly that we do not think causal relationships can be ascribed to our results.

p. 5 line 12: it isn't clear what is meant by a "real" tax. Is this just a tax that has already taken effect? If so, so state. (See also top of p. 12)

By 'real' we meant definite, rather than hypothetical. We have clarified this throughout. Page 5, par 3 and 4; page 13 par 1 and 3.

p. 5, line 39: The questions specifically about the SDIL could be asked only of the UK participants. Were similar questions asked of participants in other countries? If so, are any comparisons possible?

Similar questions were not asked of participants in other countries, precluding comparable analyses. We have clarified this: "Data from countries other than the UK were not included as comparable questions on support for, and perceived effectiveness of, the SDIL were not asked of participants from these countries." Page 7, par 2.

In Table 2, I don't see the value in reporting both the yes and no percentages for binary questions. They seem always to sum to 100%. However, it might be useful to report both the raw percentages and the percentages adjusted for the survey weights.

We have removed these 'extraneous' lines from Table 2 as suggested. As we do not believe the raw percentages represent interpretable data, we have opted not to include these.

In Table 3 I would rather not see the n and (%) for the binary variables, because the methods for this calculation are not reported and anyway, the adjusted odds-ratio is what we're here for.

We have removed these figures from Table 3 as requested. Note that Word does not mark these deletions as changes.

In Table 3, I worry that there is some spurious significance caused by the relatively large number of variables included, without strong a priori reasons to distinguish expected causal relationships. I would suggest using a Bonferroni correction to the statistical significance. That will result in fewer variables being significant, but a good deal more confidence in the ones that are significant.

Clear justification for the variables included in the analyses is already provided in the methods. We have clarified that these were documented a priori in our protocol. Page 6, par 1.

It is clearly stated in the bullet point limitations at the start of the manuscript that we do not feel the data is strong enough to support causal interpretation. This is also stated in the discussion.

We do not propose that any of the relationships found are causal, particularly given the cross-sectional nature of the survey. We have clarified this as a particular limitation, although not that the reviewer also asked us to drop the bullet point limitation about

In the discussion section, one reason these results may differ from others is the large sample size here, coupled with the use of population weights. It is possible that some of the previous results may have come from smaller and more selected samples. If so, this is a point worth making in some detail.

Whilst a larger sample size can increase precision of estimates, it is unlikely to change them, per se. However, we have acknowledged that the population weighting may be responsible for some of the differences seen compared to previously: "Higher support for, and perceived effectiveness of, the SDIL here compared to previous work may also reflect cultural differences between the UK and other countries where previous data has been collected. Unlike previously, we used population weighting which increases confidence that results are population representative." Page 15, par 3.

Another possible difference is the threshold structure of the SDIL—I doubt that this is really driving the differences, but if the authors' expertise can shed any light on this issue it would be helpful.

We have acknowledged this point in the discussion: "Finally, it is possible that the unique design and framing of the SDIL makes it more acceptable and increases perceived effectiveness compared to previous taxes proposed to research participants." Page 15, par 3.

In response to Reviewer #2

1. I suggest that the authors relate to tax on tobacco - an area which 'real-tax' was studied. This should be added to both Introduction and Discussion sections.

We have included reference to taxes on tobacco in both the introduction and discussion. Page 4 par 3; page 14, par 4.

2. Methods section - Please provide the weights that were used throughout the analysis and refer to their source. Also, add information on the software that was used to analyse the data, and add the level of significance.

It is stated on page 9, par 1 that: "Data were weighted with post-stratification sample weights constructed using population estimates from the UK census based on age group, sex and region."

We have clarified that: "Data were analysed using R version 3.3.1." Page 9, par 3.

3. Discussion - I am missing a discussion over the notable result that respondents with young children were less likely to support the tax. If we (the society in whole) aim to improve dietary habits of youngsters, it might be difficult if the parents have a negative attitude towards the tax. I would relate to this point also in the Conclusions or Implications i.e. disseminate information to the public on this regulatory measure.

Thank you for encouraging us to consider this point further. We have included a further discussion point that: "It is somewhat surprising that those with children under the age of 18 years were less

supportive of the SDIL than those without. The SDIL was particularly framed in terms of potential benefits to children. If one's own consumption is likely to influence support for the SDIL, then parents' support for the SDIL may also be influenced by their children's consumption. If children are greater consumers of sugary drinks, then this may explain why parents with children under the age of 18 years were less supportive." Page 15, par 4.

We have noted in the conclusion: "...although those with dependent children were less likely to support the SDIL" Page 16, par 3.

In response to the editorial comments

Authors must include a statement in the methods section of the manuscript under the sub-heading 'Patient and Public Involvement'. This should provide a brief response to the following questions: How was the development of the research question and outcome measures informed by patients' priorities, experience, and preferences? How did you involve patients in the design of this study? Were patients involved in the recruitment to and conduct of the study? How will the results be disseminated to study participants? For randomised controlled trials, was the burden of the intervention assessed by patients themselves? Patient advisers should also be thanked in the contributorship statement/acknowledgements. If patients and or public were not involved please state this.

We have included this subheading as requested. We have clarified that: "Patients and the public were not involved in design, conduct, analysis or interpretation of the study." Page 9, par 3.

VERSION 2 – REVIEW

REVIEWER	Frederick Zimmerman UCLA, USA
REVIEW RETURNED	04-Dec-2018

GENERAL COMMENTS	The authors have done a good job addressing all the concerns of the reviewers.
--

REVIEWER	Orly Tamir Israeli Center for Diabetes Research and Policy, Gertner Institute, Israel
REVIEW RETURNED	10-Dec-2018

GENERAL COMMENTS	My comments were properly addressed.
--------------------------------------